# Explainable Reinforcement Learning via Model Transforms

**Mira Finkelstein**[1], **Lucy Liu**[2], **Nitsan Levy Schlot**[1], **Yoav Kolumbus**[1],
**David C. Parkes**[2,3], **Jeffrey S. Rosenschein**[1] and **Sarah Keren**[4]

[1] The Hebrew University of Jerusalem, Benin School of Computer Science and Engineering
[2] Harvard University, School of Engineering and Applied Sciences
[3] DeepMind
[4] Technion - Israel Institute of Technology, Taub Faculty of Computer Science

## Abstract

Understanding emerging behaviors of reinforcement learning (RL) agents may
be difficult since such agents are often trained in complex environments using
highly complex decision making procedures. This has given rise to a variety of
approaches to *explainability* in RL that aim to reconcile discrepancies that may
arise between the behavior of an agent and the behavior that is anticipated by an
observer. Most recent approaches have relied either on domain knowledge, that
may not always be available, on an analysis of the agent's policy, or on an analysis
of specific elements of the underlying environment, typically modeled as a Markov
Decision Process (MDP). Our key claim is that even if the underlying model is not
fully known (e.g., the transition probabilities have not been accurately learned) or
is not maintained by the agent (i.e., when using model-free methods), the model
can nevertheless be exploited to automatically generate explanations. For this
purpose, we suggest using formal MDP abstractions and transforms, previously
used in the literature for expediting the search for optimal policies, to automatically
produce explanations. Since such transforms are typically based on a symbolic
representation of the environment, they can provide meaningful explanations for
gaps between the anticipated and actual agent behavior. We formally define
the explainability problem, suggest a class of transforms that can be used for
explaining emergent behaviors, and suggest methods that enable efficient search
for an explanation. We demonstrate the approach on a set of standard benchmarks.

## 1 Introduction

The performance-transparency trade-off is a major challenge with many artificial intelligence (AI)
methods: as the inner workings of an agent's decision making procedure increases in complexity,
it becomes more powerful, but the agent's decisions become harder to understand. Accordingly,
interest in *explainable AI* and the development of transparent, interpretable, AI models has increased
rapidly in recent years [1]. This increase in complexity is particularly prevalent in *reinforcement
learning* (RL) and *deep reinforcement learning* (DRL), where an agent autonomously learns how
to operate in its environment. While RL has been successfully applied to solve many challenging
tasks, including traffic control [2], robotic motion planning [3], and board games [4], it is increasingly
challenging to explain the behavior of RL agents, especially when they do not operate as anticipated.
To allow humans to collaborate effectively with RL-based AI systems and increase their usability,
it is therefore important to develop automated methods for reasoning about and explaining agent
behaviors.

36th Conference on Neural Information Processing Systems (NeurIPS 2022).

While there has been recent work on explainability of DRL (see [5] for a recent survey), most of these methods either rely on domain knowledge, which may not be available, or involve post-processing the policy learned by the agent (e.g., by reasoning about the structure of the underlying neural network [6]). Moreover, most existing methods for explainability do not fully exploit the formal model that is assumed to represent the underlying environment, typically a Markov Decision Process (MDP) [7], and analyze instead one chosen element of the model (e.g., the reward function [8]).

We focus on RL settings in which the model of the underlying environment may be partially known, i.e., the state space and action space are specified, but the transition probabilities and reward function are not fully known. This is common to many RL settings in which the action and state spaces are typically known but the agent must learn the reward function and transition probabilities, either explicitly as in model-based RL or implicitly as when learning to optimize its behavior in model-free RL. For example, in a robotic setting, the agent may have some representation of the state features (e.g., the location of objects) and of the actions it can perform (e.g., picking up an object), but not know its reward function or the probabilities of action outcomes.

Our key claim is that even if the underlying model is not fully known (or not explicitly learned), it can nevertheless be used to automatically produce meaningful explanations for the agent's behavior, i.e., even if the agent is using a model-free method, the partial model can be manipulated using a model-based analysis to produce explanations. Specifically, we suggest producing explanations by searching for a set of formal abstractions and transforms that when applied to the (possibly incomplete or approximate) MDP representation will yield a behavior that is aligned with an observer's expectations. For this purpose, we exploit the rich body of literature that offers MDP transforms [9, 10, 11, 12, 13, 14] that manipulate different elements of the model by, for example, ignoring the stochastic nature of the environment, ignoring some of the effects of actions, and removing or adding constraints. While these methods have so far been used to expedite planning and learning, we use them to automatically produce explanations. That is, while for planning the benefit of using such transforms is in increasing solution efficiency, we use them to isolate features of the environment model that cause an agent to deviate from a behavior that is anticipated by an observer.

Formally, we consider an explainability setting, which we term *Reinforcement Learning Policy Explanation* (RLPE), that comprises three entities. The first entity, the *actor*, is an RL agent that seeks to maximize its accumulated reward in the environment. The second entity, the *observer*, expects the actor to behave in some way and to follow a certain policy, which may differ from the one actually adopted by the actor. We refer to this as the *anticipated policy*, and this specifies which actions an observer expects the actor to perform in some set of states.[1] The third entity, the *explainer*, has access to a (possibly partial) model of the environment, to the anticipated policy, and to a set of MDP transforms. The explainer seeks a sequence of transforms to apply to the environment such that the actor's policy in the transformed environment aligns with the observer's anticipated policy.[2]

**Example 1** *To demonstrate RLPE, consider Figure 1, which depicts a variation of the Taxi domain [15]. In this setting, the actor represents a taxi that operates in an environment with a single passenger. The taxi can move in each of the four cardinal directions, and pick up and drop off the passenger. The taxi incurs a small cost for each action it performs in the environment, and gains a high positive reward for dropping off the passenger at her destination. There are walls in the environment that the taxi cannot move through. The observer has a partial view of the environment and knows which actions the taxi can perform and how it can collect rewards. With the information available and the, possibly incorrect, assumptions she makes about the actor's reasoning, the observer anticipates that the taxi will start its behavior by moving towards the passenger. This description of the anticipated behavior over a subset of the reachable states in the environment is the* anticipated policy. *The prefix of this policy is depicted by the green arrow in the figure. However, the actual policy adopted by the actor, for which the prefix is represented by the red arrow, is to visit some other location before moving towards the passenger.*

In order to explain the actor's behavior, the explainer applies different action and state space transforms to its model of the environment. The objective is to find a transformed model in which the actor follows

---

[1]Our formalism can be extended to support cases in which the observer anticipates any one of a set of policies to be realized.

[2]In some settings, the actor and explainer may represent the same entity. We use this structure to separate the role of an actor from the attempt to explain its behavior.

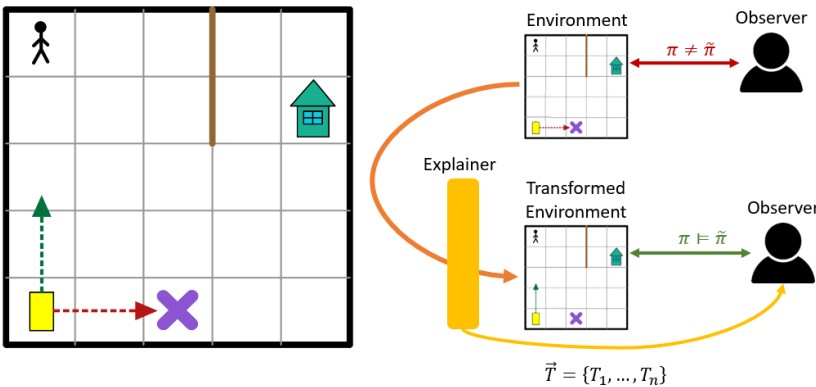

Figure 1: Reinforcement Learning Policy Explanation, example (left) and model (right). The policy prefix of the taxi (yellow square) is depicted in red and leads to the fuel station (purple mark). The observer, who is not aware of the fuel constraint, anticipates that the taxi heads towards the passenger (the anticipated policy is depicted in green). An explanation is generated by applying a transform that removes the fuel constraint.

the anticipated policy. We note that our suggested approach can produce meaningful explanations only if the explainer uses transforms that are meaningful to the observer. In our example, the explainer first applies an action transform that allows the taxi to move through walls and trains the actor in the transformed environment. Since the policy in the transformed model still does not match the anticipated policy, the explainer can infer that the reason for the discrepancy is not the fact that the observer may be unaware of the walls in the environment, and therefore this transform would not represent a meaningful explanation. As a second attempt, the explainer applies a transform that relaxes the constraint that a car needs enough fuel to be able to move, and allows the taxi to move regardless of its fuel level. After training, the actor's policy in the transformed environment aligns with the anticipated policy. This indicates the observer may not be aware of the fuel constraint, and does not expect the actor to first drive towards the gas station. This transform is consistent with the discrepancy between the anticipated and actual policies and represents a suitable explanation, as long as this constraint can be conveyed to the observer.

Beyond this illustrative example, the ability to understand the "*anticipation gap*" (the gap between the anticipated and observed behavior) is important in many applications. Examples include autonomous driving, where it is critical to know why a vehicle deviates from an anticipated course of action, medical applications, where it is crucial to explain why an AI system recommends one treatment over another, and search and rescue missions, where a robot is moving in an unknown environment with observations that are different from those of its operator and may behave in unpredictable ways.

The translation of the transform sequence that reconciles the gap between the observer and actor to natural language is beyond the scope of this work. Nevertheless, since the transforms manipulate the underlying MDP model, they incorporate the symbolic information represented by the MDP representation, and this can reasonably be expected to translate to an intuitive explanation (e.g., notifying the observer about a missing precondition in its model of an action). Thus, our approach can be used to automatically generate explanations without compromising generality. Moreover, while we used a single-agent setting to demonstrate the approach, the same ideas can apply to multi-agent settings, where the set of applicable transforms include, in addition to the transforms used for single-agent settings, transforms that deal with the multi-agent aspects of the system (e.g., shared resource constraints).

The recent interest in explainability in RL has yielded approaches that vary in the kind of questions the explanations are aimed to address and in the methods applied to find them (e.g., [16, 17, 8, 18, 19]). Ours is an example of a post-processing approach, accounting here for settings in which the observer has an anticipated behavior that is not aligned with the actual behavior, and where the objective is to find an explanation by transforming the underlying environment to one in which the agent behaves as expected.

Typically, post-hoc methods focus on a particular element of the model and investigate its effect on the agent's behavior. For example, some propose that the reward function be decomposed into an aggregation of meaningful reward types according to which actions are classified [8], or that human-designed features, such as the estimated distance to the goal, are used to represent action-value functions [18]. In other work, human-user studies have been used to extract saliency maps for RL agents in order to evaluate the relevance of features with regard to mental models, trust, and user satisfaction [19], while [6, 20] use saliency maps to produce visual explanations. Others suggest producing a summary of an agent's behavior by extracting important trajectories from simulated behaviors [21].

Our approach supports arbitrary transforms and abstractions that can be applied to the environment model and combined with any learning approach in both single- and multi-agent settings. The variety of transforms that can be used for generating explanations relies on the various methods suggested for expediting planning [13] and RL [11]. Previous work has considered an optimal planning agent in a deterministic environment and suggested learning a partial model of the environment and task, and identifying missing preconditions to explain the behavior [22]. We generalize this to stochastic environments with partially-informed RL agents and to arbitrary transforms (beyond only those that consider action preconditions).

The contributions of this work are threefold. First, we present a novel use of model transforms and abstractions, formerly mainly used for planning, to produce explanations of RL agent behaviors. Second, we present a formal definition of the Reinforcement Learning Policy Explanation (RLPE) problem and specify classes of state and action space transforms that can be used to produce explanations. Finally, we empirically demonstrate our approach on a set of standard single-agent and cooperative multi-agent RL benchmarks.

## 2 Background

Reinforcement learning (RL) deals with the problem of learning policies for sequential decision making in an environment for which the dynamics are not fully known [23]. A common assumption is that the environment can be modelled as a Markov Decision Process (MDP) [7], typically defined as a tuple $\langle S, s_0, A, R, P, \gamma \rangle$, where $S$ is a finite set of states, $s_0 \in S$ is an initial state, $A$ is a finite set of actions, $R : S \times A \times S \to \mathbb{R}$ is a Markovian and stationary reward function that specifies the reward $r(s, a, s')$ that an agent gains from transitioning from state $s$ to $s'$ by the execution of action $a$, $\mathcal{P} : \mathcal{S} \times \mathcal{A} \to \mathbb{P}[\mathcal{S}]$ is a transition function denoting a probability distribution $p(s, a, s')$ over next states $s'$ when action $a$ is executed at state $s$, and $\gamma \in [0, 1]$ is a discount factor. In this work we use *factored MDPs* [24], where each state is described via a set of random variables $X = X_1, \ldots, X_n$, and where each variable $X_i$ takes on values in some finite domain $Dom(X_i)$. A state is an assignment of a value $X_i \in Dom(X_i)$ for each variable $X_i$. To model a multi-agent setting, we use a *Markov game* [25], which generalizes the MDP by including *joint actions* $\mathcal{A} = \{A^i\}_{i=1}^n$ representing the collection of action sets $A^i z$ for each of the $n$ agents. We will hereon refer to an MDP as the model of the underlying environment, and highlight as needed the specific considerations to a Markov game.

A solution to an RL problem is either a *stochastic policy*, indicated $\pi : S \to \mathbb{P}[A]$, representing a mapping from states $s \in \mathcal{S}$ to a probability of taking an action $a$ at that state, or a *deterministic policy*, indicated $\pi : S \to A$, mapping from states to a single action. The agent's objective is to find a policy that maximizes the expected, total discounted reward.

There are a variety of approaches for solving RL problems [26, 23], these generally categorized as either *policy gradient methods*, which learn a numerical preference for executing each action, *value-based methods*, which estimate the values of state-action pairs, or *actor-critic* methods, which combine the value and policy optimization approaches. Another important distinction exists between model-based methods, where a predictive model is learned, and model-free methods, which learn a policy directly. We support this variety by assuming the algorithm that is used by the actor to compute its policy is part of our input.

## 3 MDP Transforms

We use MDP transforms to explain the behaviors of RL agents. Given a large set of possible transforms, an explanation is generated by searching for a set of transforms to apply to the environment's

model such that the actor's behavior in the modified model aligns with the observer's expectations. Since the transition from the original to the transformed environment is done by manipulating the symbolic MDP representation of the environment, the difference between the models can help the observer reason about the actor's behavior, thus providing an explanation.

In this section, we describe various transforms suggested in the literature for expediting planning and RL, and that we apply here for the purpose of explainability. We define a *transform* as any mapping $T : \mathcal{M} \to \mathcal{M}$ that can be applied to an MDP to produce another MDP. We use the term "transforms" to refer to various kinds of mappings, including "abstractions" (or "relaxations") that are typically used to simplify planning, as well as other mappings that may yield more complex environments. Moreover, the set of transforms used for explanation may modify different elements of the MDP instead of focusing on a specific element (e.g, the reward function). We provide some examples of transforms, but our framework is not restricted to particular transforms. We start by defining transforms that modify the MDP's state space.

**Definition 1 (State Mapping Function)** *A* state-mapping function $\phi : S \to S^\phi$ maps each state $s \in S$, into a state $s' \in S^\phi$. The *inverse image* $\phi^{-1}(s')$ with $s' \in S^\phi$, is the set of states in $S$ that map to $s'$ under mapping function $\phi$.

When changing the state space of an MDP, we need to account for the induced change to the other elements of the model. For this, we use a state weighting function that distributes the probabilities and rewards of the original MDP among the states in the transformed MDP.

**Definition 2 (State Weighting Function)** *[11] A* state weighting function *of a state mapping function $\phi$ is function $w : S \to [0, 1]$ where for every $\bar{s} \in S^\phi$, $\sum_{s \in \phi^{-1}(\bar{s})} w(s) = 1$.*

**Definition 3 (State-Space Transform)** *[11] Given a state mapping function $\phi$ and a state weighting function $w$, a state space transform $T_{\phi,w}$ maps an MDP $M = \langle S, s_0, A, R, P, \gamma \rangle$ to $T(M) = \langle \bar{S}, \bar{s}_0, A, \bar{R}, \bar{P}, \gamma \rangle$ where:*

- $\bar{S} = S^\phi$

- $\bar{s}_0 = \phi(s_0)$

- $\forall a \in A, \bar{R}(\bar{s}, a) = \sum_{s \in \phi^{-1}(\bar{s})} w(s) R(s, a)$

- $\forall a \in A, \bar{P}(\bar{s}, a, \bar{s}') = \sum_{s \in \phi^{-1}(\bar{s})} \sum_{s' \in \phi^{-1}(\bar{s}')} w(s) P(s, a, s')$

State-space transforms can, for example, group states together. In factored representations, this can be easily implemented by ignoring a subset of the state features. In Example 1, a state-space transform can, for example, ignore the fuel level, grouping states that share the same taxi and passenger locations.

Another family of transforms changes the action space.

**Definition 4 (Action Mapping Function)** *An* action mapping function $\psi : A \to A^\psi$ *maps every action in $A$ to an action in $A^\psi$. The* inverse image $\psi^{-1}(a')$ *for $a' \in A^\psi$, is the set of actions in $A$ that map to $a'$ under mapping function $\psi$.*

Various action space transforms have been suggested in the literature for planning with MDPs [27, 28]. Since such transforms inherently bear the MDP's symbolic meaning with regard to the environment and agent, a sequence of transforms that yields the anticipated policy can provide a suitable explanation.

As an example, even if the exact transition probabilities of actions are not fully known, it is possible to apply the *single-outcome determinization* transform, where all outcomes of an action are removed (associated with zero probability) except for one, perhaps the most likely outcome or the most desired outcome [29]. Similarly, the *all outcome determinization* transform allows a planner to choose a desired outcome, typically implemented by creating a separate deterministic action for each possible outcome of the original formulation [29, 27]. If such transforms yield the anticipated policy, this implies that the observer may not be aware of the alternative outcomes of an action, or of the stochastic nature of the environment. In settings where actions are associated with preconditions, it

is possible to apply a *precondition relaxation* transform, where a subset of the preconditions of an action are ignored [22]. For example, for MDPs represented via a factored state space, each action $a$ is associated with a set $pre(a)$ specifying the required value of a subset of its random variables. A precondition relaxation transform removes the restriction regarding these variables. Similarly, it is possible to ignore some of an action's effects, for example by applying a *delete relaxation* transform and ignoring an actions' effect on Boolean variables that are set to false [9]. As another example, a *precondition addition* transform would add preconditions to an action, perhaps those that may be considered by the observer by mistake. In all cases, if one or more transforms produce the anticipated policy, a plausible explanation is that the observer is not aware of the preconditions or effects of actions, such as in the setting we describe in regard to fuel in Example 1.

The transforms mentioned above are also applicable to multi-agent settings. In addition, we can apply multi-agent specific transforms, such as those that allow collisions between agents, or allow for more flexible communication. In a multi-agent extension of our taxi example, an observer may not be aware that taxis cannot occupy the same cell—a discrepancy that can be explained by applying a transform that ignores the constraint (precondition) that a cell needs to be empty for a taxi to be able to move into it.

## 4   Transforms as Explanations

We formalize the explainability problem as composed of three entities: an *actor*, which is an agent operating in the environment, an *observer*, which is an agent with some anticipation about the behavior of the actor, and an *explainer*, which is an agent that wishes to clarify the discrepancy between the anticipated and actual behaviors. The input to a *Reinforcement Learning Policy Explanation* (RLPE) problem includes a description of the environment (which may be inaccurate), a description of the behavior (policy) of an RL agent in the environment, the anticipated behavior an observer expects the actor to follow, and a set of possible transforms that can be applied to the environment.

**Definition 5 (RLPE Model)** *A* Reinforcement Learning Policy Explanation (RLPE) model *is defined as* $R = \langle M, A, \tilde{\pi}, T \rangle$, *where*

- $M$ *is an MDP representing the environment,*

- $A : \mathcal{M} \rightarrow \Pi$ *is the actor, which is associated with an RL algorithm that it uses to compute a policy* $\pi \in \Pi$,

- $\tilde{\pi}$ *is the anticipated policy the observer expects the actor to follow, and*

- $T : \mathcal{M} \rightarrow \mathcal{M}$ *is a finite set of transforms.*

We assume the actor is a reward-maximizing RL agent[3]. The anticipated behavior of the observer describes what the observer expects the actor to do in some subset of the reachable states[4]. Since we do not require the anticipated policy to be defined over all states, we refer to this as a *partial policy*. The settings of interest here are those in which the actual policy differs from the anticipated policy. We denote by $\mathcal{T}$ the set of all transforms. Each transform $T \in \mathcal{T}$ is associated with a mapping function for each of the MDP elements that it alters. We let $\phi_T$ and $\psi_T$ denote the state and action mapping functions, respectively (when the MDP element is not altered by the transform, the mapping represents the identity function). When a sequence of transforms is applied, we refer to the composite state and action mapping that it induces, and define this as follows.

**Definition 6 (Composite State and Action Space Function)** *Given a sequence* $\vec{T} = \langle T_1, \ldots, T_n \rangle$, $T_i \in \mathcal{T}$, *the* composite state space function *of* $\vec{T}$, *is* $\phi_{\vec{T}}(s) = \phi_{T_n} \cdot, \ldots, \cdot \phi_{T_1}(s)$. *The* composite action space function *is* $\psi_{\vec{T}}(s) = \psi_{T_n} \cdot, \ldots, \cdot \psi_{T_1}(s)$.

The explainer seeks a sequence of transforms that produce an environment where the actor follows a policy that corresponds to the observer's anticipated policy. Formally, we seek a transformed environment where the actor's policy *satisfies* the anticipated policy, i.e., for every state-action

---

[3]For the multi-agent case, instead of a single agent we have a group of agents. All other elements are unchanged.

[4]The model can be straightforwardly extended to support a set of possible anticipated policies.

pair in the anticipated policy, the corresponding state in the transformed model is mapped to its corresponding action. Given a policy $\pi$, we let $\mathbb{S}(\pi)$ represent the set of states for which the policy is defined.

**Definition 7 (Policy Satisfaction)** *Given a partial policy $\pi$ defined over MDP $M = \langle S, s_0, A, R, P, \gamma \rangle$, a partial policy $\pi'$ defined over MDP $M' = \langle S', s_0', A', R', P', \gamma' \rangle$, a state mapping function $\phi : S \to S'$, and an action mapping function $\psi : A \to A'$, $\pi'$ satisfies $\pi$, denoted $\pi' \models \pi$, if for every $s \in \mathbb{S}(\pi)$, we have $\phi(s) \in \mathbb{S}(\pi')$ and $\psi(\pi(s)) = \pi'(\phi(s))$.*

Intuitively, policy $\pi'$ satisfies $\pi$ if they agree on the agent's selected action on all states for which $\pi$ is defined. We note that our definition above is suitable only if $\pi(s)$ and $\pi'(\phi(s))$ are well-defined, i.e., if the policies are deterministic or, if they are stochastic, a deterministic mapping from states to actions is given (e.g., selecting the maximum probability action).

Clearly, for any two policies, there exist state and action mappings that can be applied to cause any policy to satisfy another policy. In order to produce valuable explanations, the input needs to include suitable transforms, i.e., transforms that change the environment in a way that highlights the elements of the model that cause unanticipated behaviors. In addition, and inspired by the notion of a *Minimal Sufficient Explanation* [8], we want to minimize the change that is applied to the environment. Intuitively, the more similar the original and transformed MDPs are, the better the explanation. We therefore assume the input to an RLPE problem includes some distance metric, $d : \mathcal{M} \times \mathcal{M} \to \mathbb{R}_+$, between a pair of MDPs [30]. In our evaluation, the distance represents the number of *atomic changes* that change a single element of the MDP (see the supplementary material for a description of several other distance metrics from the literature).

The objective of the explainer is to find a sequence of transforms that yield an MDP $M'$ such that the actor's policy in $M'$ satisfies $\tilde{\pi}$. Among the sequences that meet this objective, we are interested in sequences that minimize the distance between the original and the transformed MDP. Formally:

**Definition 8** *[RLPE Problem] Given a RLPE model $R$ and a metric function $d : \mathcal{M} \times \mathcal{M} \to \mathbb{R}_+$, an* RLPE problem *seeks a transform sequence $\vec{T} = \langle T_1, \ldots, T_n \rangle$, $T_i \in T$, s.t.*

1. *the actor's policy $\pi'$ in $\vec{T}(M)$ satisfies $\tilde{\pi}$, i.e, $\pi' \models \tilde{\pi}$, and*

2. *among the sequences that satisfy (1.), $\vec{T}$ minimizes the distance $d(M, \vec{T}(M))$.*

## 5  Finding Explanations

In an RLPE setting, the explainer has access to a set of transforms, but does not know *a priori* which transform sequence will produce meaningful explanations. This means that the explainer may need to consider a large set of possible transform sequences. This makes a naive approach impractical, as the number of transform combinations is exponential in $|T|$.

To address this computational challenge, we offer several approaches for expediting the search. Inspired by the search for an optimal MDP redesign in [31], a basic approach is a Dijkstra-like search through the space of transform sequences. Assuming a successor generator is available to provide the MDP that results from applying each transform, the search graph is constructed in the following way. The root node is the original environment. Each edge (and successor node) appends a single transform to the sequence applied to the parent node, where the edge weight represents the distance between the adjacent MDPs according to the distance measure $d$. For each explored node we examine whether the actor's policy in the transformed MDP satisfies the anticipated policy. The search continues until such a model is found, or until there are no more nodes to explore. The result is a transform sequence that represents an explanation. This approach is depicted in Figure 2, where the top of the figure depicts the search in the transform space and the lower part depicts the MDPs corresponding to each sequence.

The suggested approach is guaranteed to return an optimal (minimum distance) solution under the assumption that the distance is *additive* and *monotonic* with respect to the transforms in $T$, in that a transform cannot decrease the distance between the resulting MDP and the original one. From a computational perspective, even though in the worst case this approach covers all the possible sequences, in practice it may find solutions quickly. In addition, in cases where the transforms are

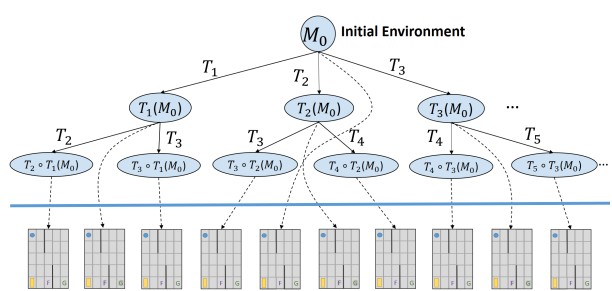

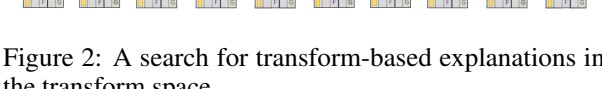

Figure 2: A search for transform-based explanations in the transform space.

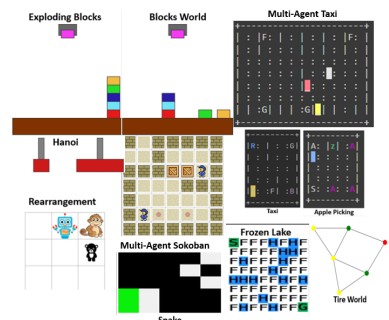

Figure 3: Evaluation domains including single-agent and multi-agent settings.

*independent*, in that their order of application does not affect the result, it is possible to expedite the search by maintaining a closed list that avoids the re-computation of examined permutations. The depth of the search can also be bounded by a predefined fixed number of transforms.

In spite of these computational improvements, the above solutions require learning from scratch an actor's policy in the transformed environment for each explored node. One way to avoid this is by preserving the agent's policy in a given environment and using it for bootstrapping re-training in the transformed environment. Another way to expedite the search is to group together a set of transforms and examine whether applying the set leads to a change in the actor's policy. If this compound transform does not change the actor's policy, we avoid computing the values of the individual transforms. This approach is inspired by pattern database (PDB) search heuristics [32], as well as the relaxed modification heuristic [31]. Even though this heuristic approach compromises optimality, it can potentially reduce the computational effort in settings in which aggregation can be done efficiently, such as when transforms have parameterized representations. In our example, if allowing a taxi to move through (all) walls in a given environment does not change the actor's policy, we avoid computing the value of all individual transforms that remove a single wall. Finally, we examine the efficiency of performing a *focused policy update*: when applying a transform, instead of collecting random experiences from the environment and updating the policy for all states, we start by collecting new experiences from states that are directly affected by the transform, and then follow the propagated effect of this change. In Example 1, when removing a wall in the taxi domain, we start by collecting experiences and updating the policy of states that are near the wall, and iteratively follow the propagated effect of this change on the policy in adjacent cells.

## 6   Empirical Evaluation

The empirical evaluation was dedicated to examining the ability to produce meaningful explanations via MDP transforms and to examining the empirical efficiency of the suggested approaches for finding satisfying explanations. Each RLPE setting included a description of the underlying environment, the actual policy followed by the actor, and the anticipated policy. We describe each component below, before describing our results[5].

**Environments:** We conducted experiments with 12 different environments, including both deterministic and stochastic domains and single and multi-agent domains (see Figure 3). *Frozen Lake* [33] represents a stochastic grid navigation task, with movements in all four cardinal directions and a probability of slipping (and terminating). As demonstrated in Example 1, *Taxi* is an extension of the similar Open-AI domain (which in turn is based on [15]), with a fuel constraint that needs to be satisfied in order to move and actions that correspond to refueling the car at a gas station. *Apple-Picking* is our stochastic extension of the Taxi domain: reward is achieved only when picking up a passenger (i.e., an 'apple') and the session can terminate with some probability when an agent encounters a thorny wall. We also used seven PDDLGym domains [34]: *Sokoban, Blocks World, Towers of Hanoi, Snake, Rearrangement, Triangle Tireworld*, and *Exploding Blocks*. The PDDLGym

---

[5]Additional results and extensions can be found in the supplementary material. Our complete dataset and code can be found at **https://github.com/sarah-keren/RLPE.git**

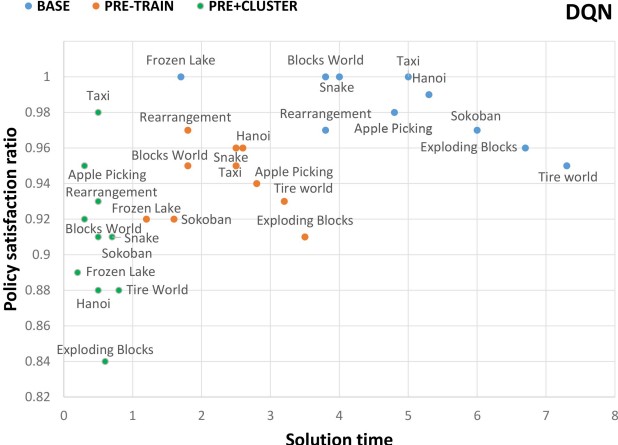

Figure 4: Single-Agent DQN [36]: Average running time for finding an explanation for each domain (x-axis), and policy satisfaction ratio, measuring the ratio of instances for which a satisfying policy was found (y-axis).

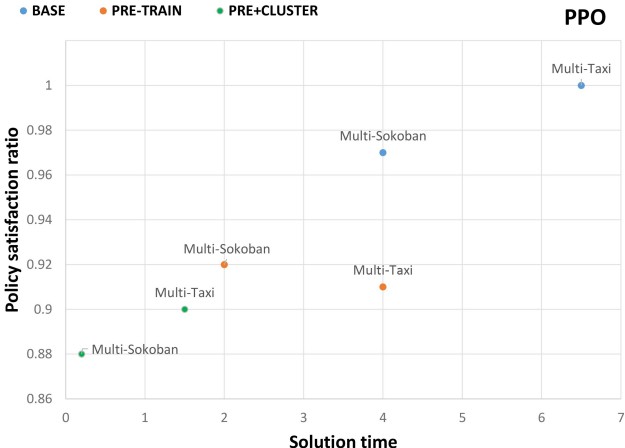

Figure 5: Multi-Agent: Comparing average running time for finding an explanation for the multi-agent domains (x-axis), and policy satisfaction ratio (y-axis). The actors are using PPO [37].

framework aligns with the OpenAI Gym interface while allowing the user to provide a model-based relational representation of the environments using PDDL [35]. This representation is not available to the actor, which operates using standard RL algorithms. For multi-agent domains, we created a two-agent Sokoban in which agents need to avoid colliding with each other and also provide a *Multi-Taxi* domain that includes multiple taxis that may collide and need to transport multiple passengers[6]. All these domains have delayed rewards and require multi-step reasoning, making them challenging for standard RL methods.

**Observer:** We considered a partially informed observer that has access to a subset of the environment features. For example, in Taxi the observer may be unaware of the fuel constraint or may not be able to see the walls. For all environments we assume the observer anticipates that the actor follows a policy that is optimal w.r.t. the observer's possibly incomplete or inaccurate model. Plans were produced using [38].

---

[6]See `https://github.com/sarah-keren/multi_taxi`

**Actor:** For the single-agent settings, we used DQN [36], CEM [39], and SARSA [23] from the keras-rl library[7], as well as Q-learning [40]. For the multi-agent domains, we used PPO [37] from keras-rl. Agents were trained for 600,000–1,000,000 episodes in each environment, with a maximum of 60 steps per episode.

**Explainer:** We used five paramterized transform types: *state space reduction* [29], *likely outcome relaxation* [29], *precondition relaxation* [22], *all outcome determinization* (for stochastic domains) [41], and *delete relaxation* [9]. Grounding (i.e., the instantiation of the parameterized representations) was performed automatically for each transform for all environments in which it is applicable. Each grounded transform modifies a single action or variable. For the Frozen Lake, Taxi, and Apple Picking domains, where the dynamics are not defined explicitly, we first learn the transition matrix to generate the *precondition relaxation* transform.

We used three methods for searching for explanations. *BASE* is a Dijkstra search, *PRE-TRAIN* is a Dijkstra search using the learned policy in a given environment to bootstrap learning in the modified environment, and with a focused policy update to avoid iteratively updating the entire policy. *PRE+CLUSTER* extends PRE-TRAIN by computing values of groups of transforms (e.g., applying the delete relaxation to multiple actions) and using them to prune individual transforms for which the superset did not change the ratio of states for which the anticipated policy is satisfied. Experiments were run on a cluster using six CPUs, each with four cores and 16GB RAM. We limited the depth of the search tree to three.

**Results:** To assess the ability to produce explanations using environment transforms, we measured the *satisfaction ratio* of each transform sequence. This measure is defined as the fraction of states for which the anticipated policy and actor policy agree among all states for which the anticipated policy is defined, i.e., the number of states $s \in \mathbb{S}(\pi)$ for which $\phi(s) \in \mathbb{S}(\pi')$ and $\psi(\pi(s)) = \pi'(\phi(s))$. For distance measure $d$, we used the length of the explanation, i.e., the number of atomic transforms (each changing a single element of the MDP) that were applied.

Figure 4 gives the results achieved by each method for the single-agent domains and with an actor that uses DQN. Figure 5 gives the results for the multi-agent settings, with PPO used by the agents. Each plot represents, for each domain and each method, the average computation time for finding an explanation (x axis) and the average satisfaction ratio (y axis), i.e., the average ratio of the expected policy that was satisfied before the search exhausted the computational resources. Results for the single agent domains show that while BASE achieves the highest satisfaction ratio (which is to be expected from an optimal algorithm), its computation time is much higher, requiring more than 7x the time of PRE+CLUSTER in Triangle Tireworld. In contrast, PRE+CLUSTER outperforms all other methods in terms of computation time, still with 84% success in the worst case domain, and with a maximum average variance of 0.03 over the different domains. The results are similar for the multi-agent settings, where the PRE+CLUSTER approach achieved best run time results on both domains while compromising the policy satisfaction rate by up to 10%.

# 7 Conclusion

We introduced a new framework for explainability in RL based on generating explanations through the use of formal model transforms, which have previously been primarily used for planning. The empirical evaluation on a set of single and multi-agent RL benchmarks illustrates the efficiency of the approach for finding explanations among a large set of transforms.

Possible extensions include integrating human users or models of human reasoning into the process of generating anticipated policies and in the process of evaluating the quality of the explanations generated by our methods. In addition, while this work uses a restrictive satisfaction relation that requires a full match between the anticipated policy and the actor's behavior in discrete domains, it may be useful to account for continuous domains and to use more flexible evaluation metrics for satisfaction that allow, for example, finding transforms that get as close as possible to the anticipated policy. Finally, our current account of multi-agent settings focuses on fully cooperative settings and it would be interesting to extend this framework to account for adversarial domains.

---

[7]https://github.com/keras-rl/keras-rl

## 8 Acknowledgments

This research has been partly funded by Israel Science Foundation grant #1340/18 and by the European Research Council (ERC) under the European Union's Horizon 2020 Research and Innovation Programme (grant agreement no. 740282).

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
