# Explainable Reinforcement Learning via Model Transforms:
## *Supplementary Material*

**Mira Finkelstein**[1]**, Lucy Liu**[2]**, Nitsan Levy Schlot**[1]**, Yoav Kolumbus**[1]**,
David C. Parkes**[2,3]**, Jeffrey S. Rosenschein**[1] **and Sarah Keren**[4]
[1] The Hebrew University of Jerusalem, Benin School of Computer Science and Engineering
[2] Harvard University, School of Engineering and Applied Sciences
[3] DeepMind
[4] Technion - Israel Institute of Technology, Taub Faculty of Computer Science

We present here additional details on our methods and additional results from our empirical evaluation. Our repository[1] contains our source code and dataset as well as a google collab notebook for running an RLPE example.

## 1   MDP Distance Measures

We aim to find a *minimal* transform sequence that yields the anticipated policy. For this purpose, we need to define a distance measure between two MDPs. A straightforward way to measure distance between a pair of MDPs $M$ and $M'$, which we use in our evaluations, is to count the number of transforms that are applied to the original MDP $M$ to produce the new MDP $M'$. However, this definition is meaningful only with *atomic transforms* that change a single element of the MDP (e.g., changing the transition probabilities of a single action). The literature is rich with a variety of other measures [1, 2, 3]. For example, in [1] the distance between two MDPs is calculated by computing the accumulated distance between every state in $M$ and its corresponding state in $M'$. In constrast, [3] use a representation of an MDP as a bipartite graph of state and action vertices and define state and action distances between two vertices recursively according to the similarity between their neighbors in the graph. They show the advantage of this method in quantifying structural similarities compared with the bi-simulation approach of [4]. The method is designed for measuring similarities within the same MDP, but is applicable in our context to measure distances between an original and a transformed MDP, as long as mappings between the states and actions of the two models are given.

## 2   Additional Results

Figures 2, 3, and 4 present the results for the SARSA, CEM, and Q-LEARNING, respectively. Each plot represents for each domain the average computation time for finding an explanation (x axis) and the average satisfaction ratio (y axis), i.e., the average ratio of the expected policy that was satisfied before the search exhausted the computational resources. As for the results presented in the main paper for DQN and PPO, PRE+CLUSTER achieved the best results in terms of solution time.

---

[1]**https://github.com/sarah-keren/RLPE.git**

36th Conference on Neural Information Processing Systems (NeurIPS 2022).

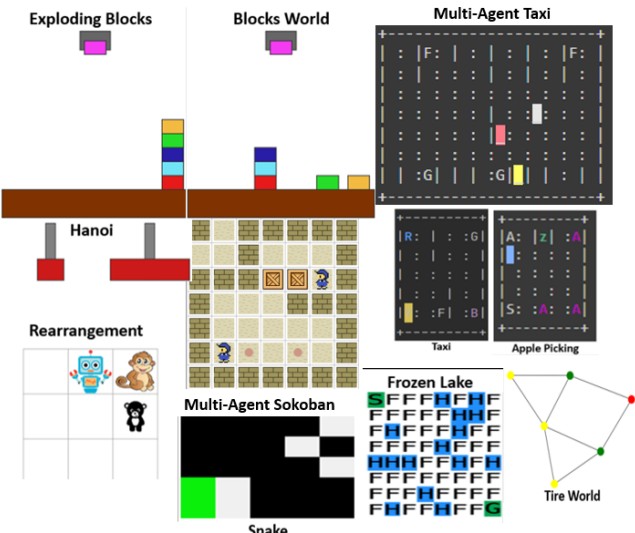

Figure 1: Evaluation domains including single-agent and multi-agent settings.

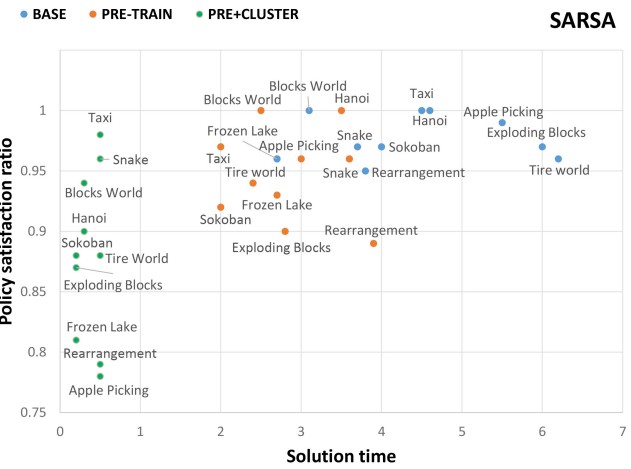

Figure 2: Policy satisfaction ratio and solution time for SARSA.

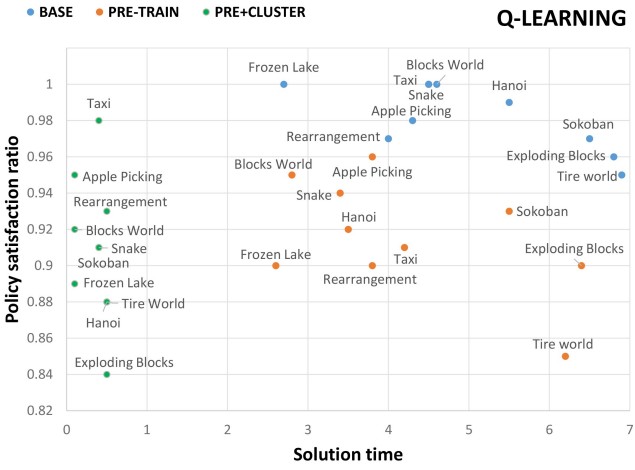

Figure 3: Policy satisfaction ratio and solution time for Q-LEARNING.

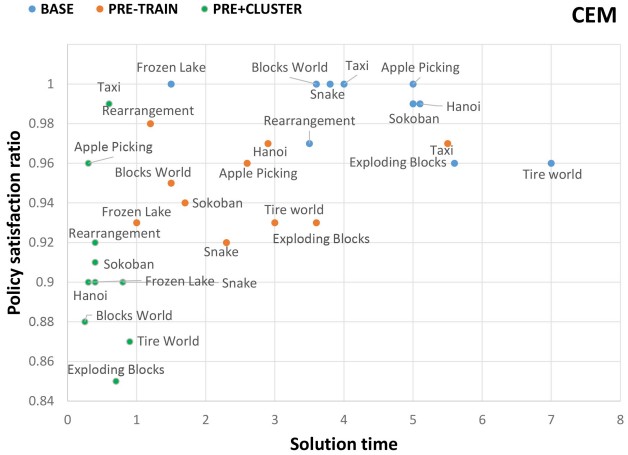

Figure 4: Policy satisfaction ratio and solution time for CEM.