# OpenReview forum: "Explainable Reinforcement Learning via Model Transforms"
_NeurIPS.cc/2022/Conference — NeurIPS 2022 Accept_

### Official Review · Reviewer_5vMh · 2022-06-29

**Rating:** 4
**Confidence:** 4
**Soundness:** 2 fair
**Presentation:** 3 good
**Contribution:** 3 good

**Summary:**

This work proposes an explainable Reinforcement Learning framework, which seeks to explain the discrepancies between the learned "actor" policy and an anticipated "observer" policy of a partially informed observer. This is done automatically by (an "explainer") searching
 over an available set of (state and action) transforms and selecting (a composition) of transforms that tweak elements in the state of the policy (producing counterfactual states), such that the (re-learned) policy in the transformed MDP aligns with the expected policy. Since the authors work on a symbolic (relational) environments, the transforms can given an indication of the key elements that had previously been overlooked by the partially informed observer (which also explains the agent's behavior, hence explainable RL). The authors do an excellent job in formalizing the definitions and explaining the intuitions. Several heuristics are proposed to perform quicker searches over the set of transforms to reach perfect alignment while making minimal changes to the states/actions. The proposed framework is shown to be applicable to a variety of domains including that of single and multi-agent RL environments, and in stochastic settings.

**Questions:**

1. lines 82:94 - In both examples, the reason behind the anticipation gap is due to the crucial information missing in the the observer's anticipated policy. How about the other way round: the actor's policy overlooking some crucial information. In other words, is the actor's policy always optimal? If not, the authors should clarify this just to avoid confusion.

2. line 106-108: I am not fully convinced with this statement. Anticipated behavior needs some domain knowledge, no matter how general it is. Moreover, I believe that the more general an anticipated behavior is, the more number of transforms would be required to align both the policies (hence, more retraining in the transformed environments), which in some complex environments might be infeasible.

3. line 185: Do these mapping functions modify multiple random variables X_i of the state or just one?

4. Line 327-328: I failed to understand how the authors can tell this without training the policy in the transformed environment?

5. Is exhaustive search used for all methods: Base, pre-train, pre-cluster? Or do you prune the transforms after a certain distance is reached?

Minor Comments:
1. To my understanding, there can be different definitions of explainability: for instance, an observer might not have any anticipations, however he/she might be interested in understanding "how" (or even "why") [1] a particular decision was arrived at. In this work, the authors focus on setups where an observer already has some expected behavior in mind. It would be good if the authors contrast this definition with those of others, or explicitly state that this is just one form of explainability they are dealing with.

2. lines 252-254: Might be good to mention this upfront to avoid confusion. As a reader, I would definitely be in suspense otherwise.

3. Scatter plots: The points should be made bigger and more distinct. Also please make the captions more detailed (like define what X-axis and Y-axis units are). As a reader, it is inconvenient to search for such details in the text.

4. Line 374: We also measured the length of the explanation (i.e. the number of atomic ... ): I couldn't find this result in the main paper. It possible I may have missed it.

line 61: "is comprised of" -> comprises
line 147-148: "where the set of states" -> where each state
line 198: s \in \phi_{-1}(\bar{s})
line 237: "as comprised of" -> as composed of

[1] Neural Logic Reinforcement Learning, Zhengyao Jiang, Shan Luo Proceedings of the 36th International Conference on Machine

**Limitations:**

The authors do propose some limitations and future work to extend their framework in the Conclusion Section. However, to my understanding, there is a key limitation that may have been overlooked:  The authors consider a Relational (symbolic) domain, hence a discretized state with limited values. In continuous domains, with the introduction of feature extraction networks, the number of elements in a state (random variables) and its corresponding values would increase, thus increasing the set of transforms required to align the two policies. Will this render the search for transforms infeasible? I understand that this might not be withing the scope of this work (since it is in the preliminary stages), however it might be good to just comment on this for future developments in this direction.

**Strengths And Weaknesses:**

Strengths:
1. This work raises valid questions on explainability in (Deep) RL, which is of great significance, given the black-box nature of the policies and their usage especially in safety critical domains like autonomous driving and treatment recommendation.
2. The authors have done an excellent job in writing the paper. I thoroughly enjoyed going through the content. The explanations and intuitions are very clear.
3. The proposed framework seems to be novel and is also generally applicable (agnostic of the RL framework). Empirical evaluations are performed on both single-agent and multi-agent environments with added stochasticity.
4. The authors have submitted code for reproducibility (although I did not run the code).

Weaknesses:
1. Results: It is rather unfortunate that the authors chose to devote just half a page to the Results sections (with just one figure) and pushed the rest of the (significant and interesting) results to the appendix. The authors have devoted a lot of space to explain things that might not be required. For instance, Figure 2 is unnecessary and can be substituted with results from the Appendix. Also I would prefer detailed explanations like those in Appendix 4 in the main paper.
2. Proposed Framework: Although the authors state that they use a RL-setting where the transition function is unknown, some action transforms that they propose work on precondition or postcondition (action effect) relaxation which is readily available in planning domains but NOT in RL domains and must be learned by the policy. Moreover, the experiments only demonstrate explainability on a limited set of transforms like all-outcome determinization (see Appendix 4) and ignore the rest.
3. Comparisons with prior work: The authors state in lines 126-130 that the work by Sreedharan et. al., 2020 is similar to theirs in deterministic environments. However, there are not comparisons presented for the deterministic experiments. I wonder why.
4. The first and the second contribution seem to go hand-in-hand: To use model transforms in the explainable RL setup, one needs to define it formally.
5. Details from the experimental section (like comparisons with previous work, model details, algorithm details) are either vaguely mentioned or are pushed to the Appendix: See Questions 3,4,5

Overall, the paper would benefit a lot from a rewrite, especially the Experiments and the Results sections.

---

> ### Author Response · Authors · 2022-08-02
> **Author Response to Reviewer 5vMh**
>
> We thank the reviewer for highlighting elements of the paper that merit further clarification. Questions raised by the reviewer are addressed below.
>
> - Proposed Framework: In our experiments, we used 5 parameterized transforms. The all-outcome deterministation is only highlighted in Appendix 4 as the leading approach. The run-time computations (including those in section 4) include all 5 transforms. Indeed, as stated by the reviewer, and is now emphasized, not all transforms are applicable in all domains. Nevertheless, we show how action effects can be learned in settings in which they are not given as input.
>
> - Comparisons with prior work: In Sreedhanan et al, the only transform that is used is the one that identifies missing preconditions (e.g., the missing fuel constraint in our example). Since such transforms are included in our evaluation, there is a comparison between the approaches in deterministic domains that are included in the paper. We highlight this important point in the current version (lines 125-128).
>
> -  Q1 anticipation gap: Our framework can support gaps in both directions. We thank the reviewer for highlighting that this important point is not sufficiently clarified, and is now highlighted in the current version (lines 53-56, 177-180, 226-227)
>
> - Q2 The intention of the statement in lines 106-108 is to highlight that the observer can use off-the-shelf transforms among the rich variety suggested in the literature. This is not to say that the explainer need not use domain knowledge to distinguish between applicable and non-applicable transforms. We have clarified this in the current version (e.g., 275-277)
>
> - Q3 According to the definition by Li, Walsh and Littman, the mapping functions can modify multiple variables. However, to maintain value in terms of explainability, it may be better to modify each variable separately, which is the approach we highlight and use in the paper.
>
> - Q4 Line 327-8: We are assuming the reviewer is referring to our heuristic estimation of grouping transforms to examine their effect. Since this is a heuristic approach, we compromise optimality but improve performance. This is shown in Figure 4 and explained in lines 378-386 , but is now further clarified in the text (line 325).
>
> - Q5 Search resources: the search is capped at a fixed and pre-chosen depth.
>
> - Per the reviewers advise, extending our framework to continuous domains  is now mentioned as an interesting avenue for future research.
>
> - All additional comments will be addressed in the final version.

---

> > ### Comment · Reviewer_5vMh · 2022-08-04
> > **Reviewer 5vMh Response to Rebuttal**
> >
> > Thank you for addressing the questions. While I am satisfied with most of them, I still have some further questions pertaining to major issues raised in my earlier comments:
> >
> > Regarding the anticipation gap: In a nutshell, I am still not convinced with the idea of having a set of (minimum) transforms to align a learned policy (here actor's) with a general anticipated policy through the setup highlighted in the paper. While the intuition sounds superficially correct, there is no way to determine if a minimum set of transforms can actually provide the correct explanation for the agent's behavior. From an optimization perspective, if the actor's policy is trapped in a local optimum, introducing/removing something in/from the state-space (like an object that provides a sub-reward) could lead to drastic changes and align it with the observer's policy. However, that tweak (or transform) may not necessarily be a reasonable (or perhaps even a correct) explanation for the initial actor's policy. Conversely, if the actor's policy is already aligned with the observer's policy, the minimum set of transforms is the Identity transform, and as such doesn't provide any explanation for the agent's behavior.

---

> > > ### Comment · Reviewer_bxZ3 · 2022-08-04
> > > **Re: Reviewer 5vMh Response to Rebuttal**
> > >
> > > The reviewer made a valid point in this statement "Conversely, if the actor's policy is already aligned with the observer's policy, the minimum set of transforms is the Identity transform, and as such doesn't provide any explanation for the agent's behavior.", but this seems to be the topic for another paper. The point of this paper is to provide explanations when the policy is not coherent with observer's expectations. I am not entirely sure if we should expect the authors to address this point in this paper.
> > >
> > > Re: the suboptimal policy. I think that there is nothing incorrect with this method with respect to the suboptimal policies. If the policy is suboptimal, the methods proposed in this paper will thus explain why the policy was suboptimal. This is perhaps another use case for this method, and I would see it a strength of the paper. Don't you think so?

---

### Official Review · Reviewer_bxZ3 · 2022-07-07

**Rating:** 8
**Confidence:** 5
**Soundness:** 4 excellent
**Presentation:** 4 excellent
**Contribution:** 4 excellent

**Summary:**

This paper studies explanations of RL policies. The idea is quite slick. An agent executes a policy learned using RL. An observer has some expectations with respect to the behaviour of the agent (e.g. it may expect certain actions to be taken in certain states). The goal of explanations is to explain to the observer why the agent's policy does not match observer's expectations. This may explain to the observer that their expectations are incorrect since they are not compatible with the MDP. The expected behaviour may simply be unfeasible given the MDP, and the goal is to find those infeasible situations and offer them as explanations to the observer. The problem is well formulated, and the results on 12 domains show that the new method is feasible, and it can be implemented even in tasks that don't have a PDDL representation.

**Questions:**

The results are excellent, but it would be useful if the appendix, for example, showed a few examples of observer's expectations, and how they were addressed by the algorithm. I can imagine that the method works, but a few examples, perhaps one example per domain, would be very useful.

Lines 18-24 in the appendix explain how to deal with non-symbolic domains? Will the derivative vector always work? I am asking because perhaps there is a counterexample that could be mentioned or a theorem that could be cited that shows that there is no counterexample.

This is a strong piece of research and I don't have other questions. The paper is well written, and it was a pleasure to read it.

**Limitations:**

There is no negative societal impact in this work.

**Strengths And Weaknesses:**

The paper is very clear and writing if of high quality. The authors took care to introduce all the necessary concepts, and the key terms have accurate formal definitions (e.g., definition 7 is excellent).

The main idea is innovative, and it appears original. The relevant literature is presented, and discussed in a way that allows the reader to see where this paper sits in the related work.

Example 1 is very good, and it (along with the paragraph in lines 82-94) clarifies the goals.

The fact that the method is domain independent is a strength.

The fact that both an intuitive explanation and formal definitions are provided is a plus.

Section 5 is very competent. The authors explain the challenges and propose a feasible solution.

One weakness is that it would be good to see what the explanations were found on the 12 domains that were evaluated.

---

> ### Author Response · Authors · 2022-08-02
> **Author Response to Reviewer bxZ3**
>
>
> We thank the reviewer for the supportive feedback and helpful comments which will be used to improve the paper’s clarity. Specifically,  according to the reviewer's very helpful advice we will add to the final version a more in-depth account and examples of the produced explanations per domain. In addition we will highlight that the approach presented for learning action preconditions is heuristic and does not offer theoretical guarantees. Counterexamples include settings with latent variables, compound (multi-variable) preconditions etc. This will be clarified in the final version of the appendix.

---

> > ### Comment · Reviewer_bxZ3 · 2022-08-03
> > **your comments read**
> >
> > Thank you for addressing reviewers' comments and questions. I don't have any further questions to you at this stage.

---

### Official Review · Reviewer_Bq81 · 2022-07-10

**Rating:** 4
**Confidence:** 4
**Soundness:** 2 fair
**Presentation:** 3 good
**Contribution:** 2 fair

**Summary:**

The paper proposes the use of MDP transforms for autonomously producing explanations. The work also introduces and formally defines the RLPE problem and empirically demonstrates the performance of their proposed approach in single and multiagent environments.

**Questions:**

Major comments:

1. The key idea is that an observer has an anticipated policy in mind which is assumed to arise from partial knowledge of the MDP. The explanation then corresponds to an MDP transformation/sequence of MDP transformations whose solution closely matches (satisfies) the anticipated policy. Although this approach might be reasonable in some cases, it is possible that the transformation sequence found to satisfy the anticipated policy may not actually constitute a good explanation. Since there may be multiple transformation sequences whose solution would satisfy the anticipated policy, only some of these may correspond to a meaningful explanation.

2. Regarding the distance metric criterion used: - if the anticipated policy was indeed derived from a transformed MDP whose distance to the original MDP is large, then choosing a transformation sequence with a low distance measure may not be ideal (because the anticipated policy actually corresponds to a transformed MDP with a large distance to the original). It would help to include some lines discussing what motivates the inclusion of the distance metric. Wouldn’t it be better to instead consider for example the length of the transformation sequence and select ones with lower sequence lengths (as smaller lengths might mean simpler explanations)

3. How does one choose a suitable set of transformation functions? This is probably a critical choice because a poor set of transformations may still be able to match the anticipated policy, but may not lead to meaningful explanations.

Minor comments:

1. The figure captions need to be more detailed. Eg: It is not clear what the yellow rectangle/purple ‘x’ in fig 1 mean.
2. The phrasing of lines 164-165 is confusing
3. In lines 351-355, I am curious as to why A* was used – was it because it assumes the existence of some heuristic signal which might simulate an observer’s assumptions? Why not use say standard Q learning for example?
4. It would be good to see results in more complex/high dimensional environments. The chosen environments all seem relatively simple.
5. The results only report the satisfaction ratio with the transformed MDPs. Reporting the satisfaction ratios on the original environments might better indicate how much the transformation sequence improves the satisfaction ratio relative to the original environment.

**Strengths And Weaknesses:**

The article is clear and fairly well written, and the proposed approach seems novel and original. I believe the paper tries to tackle a very relevant and important problem. My main concerns with the work (as listed later) are related to the validity of the assumptions made with respect to the quality of explanations produced by the proposed approach.

---

> ### Author Response · Authors · 2022-08-02
> **Author Response to Reviewer Bq81**
>
> We thank the reviewer for the helpful comments that will help clarify the main ideas of the paper.
>
> - Q1 and Q3: As we mention in the paper (and now emphasize in lines 274-7) the quality of the transform will dictate the quality of the produced explanations and our ability to find relevant ones. This means that it is up to the user to choose meaningful transforms. To demonstrate, if the method is given the option to select the trivial transform that would 'force' the agent to behave according to the anticipated policy (by removing all other actions from the set of applicable actions in that state), this will yield a policy that "satisfies" the anticipated policy but is meaningless for explainability. That being said, the RL and automated planning literature is rich with transforms that accept as input an MDP  and can be used 'off-the-shelf' to produce meaningful explanations.
>
> - Q2: We agree with the reviewer w.r.t. the suggested distance measure, and note that this is the measure we describe in the paper and use in our experiments : i.e., we consider the length of the sequence and select explanations of minimal length. We also note that we define a general distance metric for exactly the reasons mentioned by the reviewer, allowing the user to define the distance measure that is most relevant to the setting at hand.
>
>
> - We chose a symbolic state-space search (A*) because the state space is implicitly given by the set of applicable transforms and a deterministic transition function for which the outcome is known and therefore does not need to be learned.  Extending our approach to settings in which these assumptions don't hold, requireling an RL algorithm for the search, is an interesting avenue for future work.
>
> - The domains we chose include long-term reasoning and delayed rewards and were therefore challenging for the standard RL we chose for experimentation.
>
> - The only instances we consider in our evaluations are settings in which there is a difference between the anticipated and actual policy. Therefore examining the satisfaction ratio in the original environment is always 0, making it useless to consider the satisfaction ratio relative to the original environment. We will clarify this point.

---

> > ### Comment · Reviewer_Bq81 · 2022-08-04
> > **Thanks for the response!**
> >
> > I thank the authors for their responses.
> >
> > Q1 and Q3: With regards to the point about the quality of transforms and explanations, I think this point should be mentioned upfront, and not later in the paper. I think it a major limitation of the paper. Having said that, it facilitates your novel approach. So it may be justified as long as it is declared upfront, while clearly stating the scope of the problem you are dealing with.
> > On a related note, even if a set of intuitively meaningful transforms were selected, would it be possible to end up with less meaningful/meaningless explanations? For example, I imagine it might be possible to end up with a sequence of individually meaningful transforms, which correspond to a not-so-meaningful explanation.
> >
> > Q2: Regarding minimal length explanations: Thanks for clarifying this point, I may have missed it in the submitted version. Now it makes more sense to me.
> >
> > Minor comment 4: The nature of environments (long term reasoning + delayed rewards) chosen is fine. The reason I raised the issue of small environments was to clarify whether the authors have been able to scale these approaches. For instance, I imagine the space of possible transforms may become impractically large when dealing with large state/state-action spaces.

---

> > > ### Author Response · Authors · 2022-08-04
> > > **Clarifications on limitations added**
> > >
> > > - Per the reviewer's suggestion, additional clarifications were added to the paper regarding the reliance on meaningful transforms to produce meaningful explanations (85-86 in the recently uploaded version). That being said, the authors wish to highlight the rich and ever-evolving literature on transforms and abstractions that can be used within the framework we offer (e.g.  Abel et a. ICML 2018)
> > >
> > > - A group of meaningful transforms may yield meaningless explanations - which is why we suggest to bound the search and explore explanations of increasing size.
> > >
> > > - Scale was achieved using our suggested extensions to BFS and our suggested heuristics. This compromised completeness, but substantially improved performance (see Figure 4).

---

### Official Review · Reviewer_2xom · 2022-07-12

**Rating:** 3
**Confidence:** 5
**Soundness:** 3 good
**Presentation:** 2 fair
**Contribution:** 1 poor

**Summary:**

This paper presents a new framework and algorithm for explainable RL.  Given some limited domain information and observed behavior by an actor, the authors propose to search through a space of possible model transformations in order to find a model and associated policy that matches the observed behavior.  The transformations then represent some aspect of deviation, and serve as a partial explanation for observed behavior.  The transforms are derived from RL literature on state-space aggregation; the search algorithm involves chaining them together. Each node in the search tree must be evaluated by computing a full policy for the transformed model, which can be computationally expensive, and which the authors discuss.  The paper has one set of experimental results illustrating an implementation of the ideas.

**Questions:**


- Why do we need an observer?  Why not eliminate that idea and simply state that there is an anticipated policy?

- The claim on lines 274-275 isn't obviously true to me. Can you please explain this further? (I think I understand why you might think it's "clear", and I think you might be wrong, so it would be helpful to have it fleshed out)

**Limitations:**

I think the authors could have done a better job of explaining the limitations of their method (although I don't think they tried to hide anything).

**Strengths And Weaknesses:**

Strengths:

+ The framework is novel
+ This is a reasonable problem to be working on
+ The ideas are straightforward, and the paper is clearly written
+ The experiment seems clear and helpful

Weaknesses:

- While I generally liked this paper, I consistently found myself wanting more. The paper is very "chatty" and does not do a good job of balancing exposition of the ideas (which are straightforward) with some sense of application or empirical evaluation (which is very, very thin).

- I ultimately felt the problem (explainable RL) is important, but that the proposed framework should have been more general. I mean this in two specific ways:

1) A more general characterization of the idea could be "search for a model and corresponding optimal policy that matches observed behavior". This subtly includes a host of ideas that involve /complexifying/ policies, instead of just /simplifying/ them.  For example, you discuss how removing a constraint from a symbolic planning problem might suggest that the agent was unaware of the constraint, but the opposite is also true - you could /add/ constraints to the problem,  to similarly better match behavior, that would suggest the agent is laboring under constraints that it doesn't need to.  People do this all the time!

2) I am generally dissatisfied by the approach because I feel that it can only explain the /difference/ between the baseline policy and the transformed policies. Suppose, for example, that the optimal set of transforms is the null set -- in other words, the agent is behaving as expected.  This does NOT imply to me that the agent's behavior is explainable!  In other words, this framework does not contribute any explainability of the base policy. In my mind, this also motivates the more general approach of searching for matching models.

- The framework seems a bit limited to what feels like counterfactual reasoning

- The computational cost is real.  As the space of transforms grows, the importance of searching that space effectively becomes increasingly important. You discuss how important a distance metric is, but it's unclear to me how often such a thing exists or is practical to compute.  I like the idea of somehow transferring partial solutions or bootstrapping solvers, but these seem to dance around the issue.  Ultimately, I'm lead to wonder: will this method ever be useful for real problems?

- The experiments in the paper are underhwhelming. While the single provided experiment is clear and helpful, I feel like a much stronger empirical evaluation is necessary.  I would recommend trimming the writing to make room for experiments.

---

> ### Author Response · Authors · 2022-08-02
> **Author Response to Reviewer 2xom**
>
> We thank the reviewer for the insightful and knowledgeable comments that we used to improve clarity of the paper.
>
> - As mentioned by the reviewer, the key idea of our approach is the ability to support a variety of transforms, including those that simplify *and* those that render a more complex problem (e.g. by adding constraints). We thank the reviewer for pointing out that this important point isn't sufficiently clear, and it is now highlighted in the modified version (lines 53-56, 177-180, 226-227)
>
> - Our approach does not guarantee the behavior of the actor is explainable. As we mention in the paper (and now highlight) the quality of the explanations highly depends on the set of transforms that is provided as input.
>
> - We believe our method is useful for various real world problems. While many approaches aim at making sense of the underlying DRL network, we suggest an automated way to produce potential explanations without relying on an understanding of the internal structure and implementation. In fact, the idea for this work came from discussions we had with practitioners and AI researchers that highlighted the need for such a framework.
>
> - As we state in footnote 2, we do not need an observer. The observer's role is merely to highlight the need for explainability
>
> - The statement "Clearly, for any two policies there exist state and action mappings that can be applied to cause any policy to satisfy another policy" in lines 274-5 is aimed to emphasize the importance of providing meaningful transforms as input. The trivial transform would be one that forces the agent to behave according to the anticipated policy by removing all other actions from the set of applicable actions in that state (e.g. by disallowing moving right from the initial state in our example). This will yield a policy that "satisfies" the anticipated policy but does not contribute to explainability.
>
> - As we mention on page and highlight in the current version, our approach has two key limitations: 1. explanations are only as good as the transforms that are provided as input. 2. we do not directly deal with the translation of the produced explanations to natural language and assume instead that this is something that is straightforward to accomplish.

---

### Author Response · Authors · 2022-08-02
**Addressing the comments and questions of the reviewers**

We thank all reviewers for their helpful feedback on the paper. All comments were used to improve clarity of the paper.
We uploaded a revision of the paper that includes the clarifications requested by the reviewers and which are individually addressed below.

---

> ### Comment · Reviewer_bxZ3 · 2022-08-03
> **from reviewer to other reviewers**
>
> You wrote expert reviews and you asked excellent questions. The authors tried to answer your questions. The reviewers' discussions with the authors will end in a few days. If you'd have further questions to the authors, it would be good to do that asap. This message is written by another reviewer. I am sorry for chasing up, but it would be a shame if this paper was rejected without a thoughtful consideration.

---

> > ### Author Response · Authors · 2022-08-08
> > **Any additional questions and concerns ?**
> >
> > Dear reviewers,
> > As the author-reviewer discussion period is about to end, we would like to know if you have any additional concerns or questions in light of our responses? If so, we will be happy to address them.

---

### Meta-Review · Area_Chair_o9it · 2022-08-24

**Recommendation:** Accept
**Confidence:** Less certain

**Metareview:**

This paper is about explainable AI: explaining a black-box agent's learned behavior via how it aligns with an observers anticipated behaviour

This paper was a bit polarizing with the reviewers. First let's summarize on the agreements between the reviewers. They all agreed:
+ the problem of study is interesting and important
+ the paper was well written and engaging
+ the formalism and overall approach is unique

The disagreements between the reviewers revolved around:
- sharpening the text and claims within (addressed via extensive author engagement)
- the availability of good transforms
- the meaningfullness of the explanations when the agent's behavior is degenerate or there is agreement with the observer
- the completeness of the experiments provided (limited space and focus compared to other parts of the paper)
- computational tractability of the approach (searching transforms and solving for optimal policies)
- applicability in real world settings (focus on discrete symbolic domains, computation again)

As one reviewer put it "would anyone really use this approach?". Three reviewers aligned on clear reject concluding that the paper was intriguing but there were too many loose ends and open questions left to future work. Whereas the 4th reviewer found the work to be more than enough.

The AC found both sides of the argument compelling with a slight concern that more substantive ideas were required to convince the reader that the approach is applicable to high-dimensional, messy domains like pixel-based control and robotics. Indeed, symbolic domains are great for illustration purposes, but those domains are so intuitive that behavior is often interpretable by design---afterall they are toy problems designed to highlight specific agent properties. Whereas, messy domains, like autonomous driving (operating on multiple dimensions of sensors, actuators --- all at different timescales) which the paper used as motivation are the end goal, and it remains unclear if this paper can get us there.

In the end the paper is well executed and correct so should be considered for acceptance.

**Award:**

No

---

### Decision · Program_Chairs · 2022-09-14

Accept